# Functional Validation of Osteoporosis Genetic Findings Using Small Fish Models

**DOI:** 10.3390/genes13020279

**Published:** 2022-01-30

**Authors:** Erika Kague, David Karasik

**Affiliations:** 1School of Physiology, Pharmacology and Neuroscience, Biomedical Sciences, University of Bristol, Bristol BS8 1TD, UK; erika.kague@bristol.ac.uk; 2The Musculoskeletal Genetics Laboratory, The Azrieli Faculty of Medicine, Bar-Ilan University, Safed 1311502, Israel

**Keywords:** genome-wide association study, skeletal disease, gene regulation, causal gene, zebrafish, osteoblast, osteoclast, data integration

## Abstract

The advancement of human genomics has revolutionized our understanding of the genetic architecture of many skeletal diseases, including osteoporosis. However, interpreting results from human association studies remains a challenge, since index variants often reside in non-coding regions of the genome and do not possess an obvious regulatory function. To bridge the gap between genetic association and causality, a systematic functional investigation is necessary, such as the one offered by animal models. These models enable us to identify causal mechanisms, clarify the underlying biology, and apply interventions. Over the past several decades, small teleost fishes, mostly zebrafish and medaka, have emerged as powerful systems for modeling the genetics of human diseases. Due to their amenability to genetic intervention and the highly conserved genetic and physiological features, fish have become indispensable for skeletal genomic studies. The goal of this review is to summarize the evidence supporting the utility of Zebrafish (*Danio rerio*) for accelerating our understanding of human skeletal genomics and outlining the remaining gaps in knowledge. We provide an overview of zebrafish skeletal morphophysiology and gene homology, shedding light on the advantages of human skeletal genomic exploration and validation. Knowledge of the biology underlying osteoporosis through animal models will lead to the translation into new, better and more effective therapeutic approaches.

## 1. Introduction

Osteoporosis is the most prevalent bone condition in the ageing population. It is a disease that weakens the bones, making them highly susceptible to sudden and unexpected fractures [1]. It is estimated that, worldwide, 1 in 3 women and 1 in 5 men, over the age of 50, will experience osteoporosis fractures in their remaining lifetimes. In 1990, it was projected that by 2050, the worldwide incidence of hip fractures in men would increase by 310% and 240% in women. It is estimated that by 2040 over 320 million individuals will be at high fracture risk [2]. Regardless of such alarming estimates, the molecular mechanisms of osteoporosis still remain blurry.

Genome-wide association study (GWAS) and whole genome sequencing (WGS) analyses have transformed the field of genetics of complex diseases in general, and for osteoporosis (OP) in particular. Although a range of modifiable factors contribute to age-related bone loss, including diet, exercise, medications, and comorbidities, primary OP is largely mediated by genetic factors, making it a promising field for exploration by GWAS and WGS [3].

Bone mineral density (BMD) remains the strongest predictor of fracture risk and is highly heritable [4]. Over the years, research in OP etiology has focused on BMD (and changes in BMD), clinical diagnosis of osteoporosis, and/or osteoporotic fractures. Although the majority of these studies used dual-energy X-ray absorption (DXA) scans to evaluate BMD parameters, alternative measures, most prominently by quantitative ultrasound of the calcaneus bone, are well studied [5,6]. Less conventional traits such as trabecular volume, thickness and number, periosteal expansion, and cortical volume and thickness were also studied [3]. BMD of other regions, such as jaws or metacarpals, were not studied extensively. The skull has been of recent interest, due to the possibility of enrichment and discoveries of intramembranous ossification-associated genes in the etiology of OP. In addition, given the relatively lower mechanical load on the skull, osteocytes are highly mechanosensitive, opening opportunities for the identification of genes involved in fine-tuning mechanical regulation of osteocytes [7,8]. Bone geometry and bone size/shape GWAS was also performed over the recent decades [9,10].

Heritability of OP-associated traits may have quite a range—from 10 to 80%—but is always significantly different from zero. Despite the strong heritability associated with osteoporosis, the identification of causal genes remains complex. Historically, studies in mice have added functional evidence to accompany GWAS. More recently, zebrafish have been used as alternative models to explore variants harbored in protein-coding regions. In this review, we will address current challenges faced by researchers to bridge the wide gap between association and causality in osteoporosis. In addition, we will demonstrate the power of functional studies using small teleost zebrafish to underpin the genetics of osteoporosis and the biology behind this condition that currently affects over 200 million people worldwide.

## 2. Large GWAS and WGS Have Identified Multiple Loci Associated with Human Skeletal Traits

In 2007, the first GWAS of the then commonly measured osteoporosis-related phenotypes was published [11]. Since then, hundreds of loci and SNPs with associations to osteoporosis have been identified. The collaborative study of the Genetic Factors for Osteoporosis Consortium (GEFOS) cohort resulted in the identification of 20 loci associated with BMD [12]. The results from GEFOS2 [13] provided additional associations for fracture risk traits, as did Zheng et al. later on [14] (Figure 1). In a recent review, Zhu et al. summarized the clinical use of GWAS findings in the bone field, which included the identification of risk factors, development of drug targets, and OP risk prediction [15]. We highlight current drug targets for OP, which therapeutics have been developed for (Figure 2). Inhibition of sclerostin by romozosumab is currently the most efficient therapeutic.

Studies conducted on fracture risk accumulated large sample sizes, including data from the GEFOS consortium and UK Biobank [1]. An important finding from the GWAS conducted on BMD is the consistent support of the direct correlation between loci decreasing BMD and the occurrence of fractures. A large-scale GWAS meta-analysis by Trajanoska et al. including 25 cohorts with genome-wide genotyping and fracture data identified 15 genetic determinants of fracture, such as a genetic predisposition to lower vitamin D and calcium intake, as well as clinical risk factors from comorbidities, such as diabetes and rheumatoid arthritis [1]. Using a Mendelian randomization approach, the authors reconfirmed that genetically decreased BMD was the only clinical predicting factor with an effect on fracture risk among those tested [1]. A Mendelian randomization GWAS re-analysis found an inverse correlation between the concentrations of serum parathyroid hormone, which regulates calcium absorption, and BMD [16]. These Mendelian randomization studies help to outline shared genetics with other risk factors and systemic diseases, while still substantiate BMD and fracture-specific loci and variants.

The largest GWAS to date on the genetics of osteoporosis is from the UK Biobank study, involving approximately 420,000 participants [17]. This study identified a total of 518 loci associated with estimated heel BMD, of which 301 were *new* loci (Figure 1). To date, the bone-related function for the majority of these genes is still unknown. Morris et al. performed pathway analysis for potential causal genes within 100 kbp of the top SNPs, identifying known pathways such as Wnt signaling, endochondral ossification, osteoclast and osteoblast signaling. In addition, other pathways were identified such as DNA damage response, neural crest differentiation, mesoderm commitment, TGF-β signaling, FTO obesity variant mechanism, transcription factor regulation of adipogenesis, estrogen signaling, and BMP signaling and regulation [17]. Pathway analysis demonstrates the complex function of candidate genes and reinforces the need for functional studies to clarify causality. More recently, summary data-based Mendelian randomization (SMR) analysis was implemented to investigate new genes and loci associated with BMD [18]. SMR performs an aggregation of the GWAS SNPs with data from expression quantitative trait loci (eQTL). The authors identified 12,477 SNPs that regulated 564 genes, which are associated with BMD [18]. Still, the function for many of these genes is unknown, especially whether it is relevant to bone homeostasis. Very recently, Medina-Gomez et al. performed the largest skull-BMD-GWAS meta-analysis involving 43,800 individuals, identifying 59 loci, from which four loci were novel [7]. Using functional studies in zebrafish, the authors demonstrated that skull-BMD GWAS has great potential to help uncover genes important in developmental craniofacial conditions, such as craniosynostosis [7]. Remarkably, this work showed that the majority of genes involved in skull homeostasis and intramembranous ossification also play a role in the maintenance of the remaining skeleton and are equally important to understanding the biology of osteoporosis.

### WGS to Boost the Identification of Rare Variants Associated with Human Skeletal Traits

Recently, WGS had become a tool of choice to identify rare variants with large effects, if contributing sample size is substantial. Currently, WGS studies have identified several rare mutations in *LGR4* [19] and *COL1A2* [20] associated with low BMD. Table 1 summarizes the WGS of OP-related phenotypes.

The UK BioBank (UKBB) has recently sequenced whole exomes of >200 thousand participants, which were made available to the Exome Sequencing consortium. A subset of that enormous sample (*n* = 49,960) was released to be used by the wider scientific community [22]. Among multiple autosomal loss-of-function (LOF) variants with large effects on disease traits, they found novel LOF burden (cumulative minor allele frequency (MAF) = 0.18%) in *MEPE* associated with heel bone density. To date, exome data from 200,643 UKBB enrollees are now available, which include ~10 million exonic variants. Interpreting results from human association studies requires bridging the gap between genetic association and molecular function. A systematic functional investigation is necessary to interpret significant variants/genes they regulate, to decipher the exact disease-causing genes, and the cells in which they act [23].

## 3. Genetic Mutations Associated with Rare Skeletal Diseases also Require Functional Validation and Proof of Causality

Rare skeletal disorders span a broad clinical spectrum of bone-related pathologies and are sometimes accompanied by extra-skeletal manifestations. These syndromes and disorders are highly variable, ranging from neonatal lethality to minor ailments discovered incidentally during adulthood [24,25]. Although not a focus of this review, genetic predisposition to rare skeletal disorders frequently overlaps with that of OP [26,27]. Similar to complex diseases such as OP, all genetic discoveries resulting from traditional approaches such as linkage analysis in the multiplex families or high-throughput sequencing (usually WES) technologies require translational assessment and annotation using in vitro or ex vivo bone cell work and/or in vivo knockout models, in mice or fish, to confirm disease association [28].

Related to the anti-OP treatment, atypical femur fractures (AFFs)—rare subtrochanteric or diaphyseal fractures—are regarded as side effects of bisphosphonates (BPs). Zhou et al. summarized the most recent knowledge about the genetics of AFFs [15]. AFFs had been reported in patients with different monogenic bone disorders including hypophosphatasia and osteogenesis imperfecta, pycnodysostosis, osteopetrosis, osteoporosis pseudoglioma syndrome, etc. [29]. The genes that have been implicated in AFF include those associated with monogenic bone disorders and some involved in the action of BPs, such as *GGPS1* and *ATRAID*. Thus, the poorly characterized gene, *ATRAID,* is required for alendronate inhibition of osteoclast function. In exome sequencing on AFF patients taking BPs, *ATRAID* was found to contain rare non-synonymous coding variants associated with the poor outcome of BP treatment [30]. WES was conducted on three sisters, who had a history of AFFs after long-term BP treatment for their underlying osteoporosis [31]. WES analyses identified the presence of a rare missense mutation in *GGPS1,* encoding the Geranylgeranyl Diphosphate Synthase 1 enzyme, which acts downstream of the point of bisphosphonate action. Other WES-prioritized variants, such as *CYP1A1*, were also found mutated in AFF cases [32,33]. Bisphosphonate-induced osteonecrosis of the jaw (BRONJ) is also heritable [34]. Recently, WES analyses resulted in a modest success in identifying variants associated with this adverse event [35].

## 4. Current Practice: Genomic Annotation and Establishing Causality

### 4.1. Challenges of Genomic Annotation for Coding and Non-Coding Regions

The genes regulated by SNPs can be located tens or hundreds of kilobase-pairs away and mingled among other genes, making the discovery of the actual regulated gene a tremendous challenge [36]. Intuitively, if there is only one gene at a GWAS locus, the probability of that gene being causal is higher than if there are multiple genes at the same locus. The problem is exacerbated when the locus is within a “gene-desert” region.

It was shown that the distance between the lead SNP and the neighbor gene is a strong predictor of causality, yet there are examples of causal genes at GWAS loci that lie hundreds of kilobases away from the lead SNP [37,38]. The identification of candidate genes is more straightforward for coding variants, which may directly disrupt the structure of a protein and thus could provide clues to interpreting the function of the gene [23]. Protein-coding sequences are enriched in GWAS-significant signals, and while exons constitute only ~1.5% of the human genome, they are responsible for ~10% of GWAS hits. Still, it was realized early on that among GWAS variants, only a minority fall within transcribed regions, with most of them mapping to introns (4.9% and 41.2%, respectively [39]). The overwhelming majority of the GWAS-discovered SNPs occupy non-genic portions of the genome that do not result in an obvious disruption in the coding sequence, making them challenging to interpret. Furthermore, GWAS-reported tag SNPs may merely be markers co-segregating with the causal mutation [36].

Since the majority of GWAS hits cannot be easily linked to a candidate causal gene, multiple methods and software are dedicated to the purpose of variant annotation, as depicted in recent reviews [40,41]. Of note, large-scale human-centered undertakings, such as the ENCODE Project Consortium [42] or the Roadmap Epigenomics Consortium [43] were poor on the representation of skeletal cells, including only one human osteoblast cell line. In addition, there is no established bone data in GTeX [44] to characterize human transcriptomes for a wide variety of primary tissues and cell types. Hopefully, in the near future, especially with the rise of single-cell sequencing, similar information will become readily available for human bone cell types.

With WES being increasingly applied to large population-based settings, its annotation seems more straightforward (since these SNPs are coding). The American College of Medical Genetics and Genomics and the Association for Molecular Pathology have released standards and guidelines for the interpretation of sequence variants [45]. According to these guidelines, the variants are classified from 1 = benign to 5 = highly pathogenic. Databases can follow this classification system, whereas others, such as the Human Gene Mutation Database (HGMD), use their own adaptation of functional classifications [46,47]. Thus, in the ClinVar database, the variants are characterized into class 1 (benign), 2 (likely benign), or 3 (uncertain significance) for the clinical interpretation [48]. The Combined Annotation Dependent Depletion (CADD) is a tool that uses a machine learning approach for scoring the deleteriousness of both coding and non-coding variants [49]. As follows, the definition of known pathogenic variants is ambiguous between commonly used databases. This presents challenges that we face even with the “intra-genic” variants while interpreting our findings. Finding and replicating statistical association with an SNP is not the end of the road, it is rather a mid-way to discovery. The development of methods to annotate variants within non-genic regions is a race that continues at a high and unprecedented pace [37,50].

The high complexity of the gene regulatory landscape also adds to the challenge in interpreting GWAS findings. Recent works have demonstrated that the genetic architecture of disease-associated loci may involve extensive pleiotropy [51]. GWAS variants on a common haplotype can modify the regulatory properties of several enhancers targeting multiple genes [52]. Only by extensively combining -omics, bioinformatic tools, high-throughput in vitro assays and in vivo analysis will it be possible to unravel gene causality. This is exemplified by the recent dissection of the regulatory landscape of the *FTO* locus, which has been strongly associated with the risk of obesity [37]. Sobreira et al. used 4C-sequencing to reveal long-range interactions between the obesity-associated locus and promoters of *IRX3* and *IRX5* but not those of *IRX6* or *FTO* [53]. By genetically engineering *Irx3* and *Irx5* mice, the group showed the presence of an anti-obesity phenotype in *Irx3* knockout and *Irx5* heterozygous mice. Their studies confirm that variants in multiple enhancers within *FTO* obesity-associated regions regulate the expression of multiple genes (*IRX3* and *IRX5*) in at least two obesity-relevant tissues (adipose and brain). In another study, Chesi et al. efficiently combined open chromatin landscape (Assay for Transposase-Accessible Chromatin sequencing, ATAC-seq) with chromatin interactions data (Capture C) and BMD GWAS loci to map informative SNPs [54]. SNP promoter interactions showed that overall, 46% of the GWAS loci interacted with only one promoter region, while 54% interacted with more than one promoter. Chesi et al. further tested one known BMD-GWAS locus, harboring *CPED1*, *WNT16,* and *SFRP4*, using cell cultures. They showed that knockdown of two novel genes (*ING3* and *EPDR1*), not previously associated with BMD but highlighted through their chromatin analysis, affected osteoblast and adipogenic differentiation [54]. Together, these works provide evidence that SNPs even confined to a single gene can lead to an incomplete understanding of causality. Therefore, the ability to test several genes in parallel or in combination is an attractive approach for functional studies. There is a high demand for exploring and developing alternative and more rapid functional platforms that would allow parallel approaches towards dissecting complex loci identified by GWAS.

### 4.2. Evolutionarily Conserved Coding and Non-Coding Regulatory Elements

Given the recent expansion of sequenced genomes, it became possible to intersect datasets for evolutionarily conserved genes, regulatory elements, or entire pathways with the evolution of homologous traits in a relatively unbiased fashion [55,56]. While conservation in coding DNA among vertebrates is high, conservation in non-coding DNA is less prominent. High levels of divergence are observed in UTRs, introns, and intergenic DNA [57], which often harbor trait-associated SNPs. Regarding sequence similarity, only <4% of non-coding sequences are highly conserved among mammals [58,59]. Further, less than 1% of non-coding sequences are conserved with more distant vertebrates, such as teleost [58]. In comparison, >70% of protein-coding genes and >20% of small RNA primary transcripts could be traced to the last common ancestor of tetrapod and ray-finned fish [60].

It has been postulated that cis-regulatory sequences and other non-coding elements may be a primary substrate for evolutionary divergence (reviewed by Kwon et al. [61]). Indeed, conserved non-protein-coding DNA elements (CNEs) are likely critical for the expression of the evolutionarily conserved trait [62]. In addition, ultra-conserved elements (UCNEs), defined as being at least 200 bp long and sharing 100% sequence identity between human, mouse, and rat genomes [63], are recognized.

Ancient CNEs that arose in the vertebrate ancestor usually have only a small core of sequence conservation in fish species, while mammals often exhibit much broader conserved flanks. This observation inspired Madelaine et al. to look for GWAS-originated SNPs that fall in deeply conserved CNEs, from humans down to zebrafish, that also preserve gene synteny [36]. Synteny is defined as a co-localization of genes on chromosomes of different species, therefore preserved in evolution. They consequently identified only 45 non-coding human SNPs located in deeply conserved CNEs. Interestingly, when the authors relaxed the evolutionary constraint to 300 million years between human and chicken, versus 450 million years with zebrafish, and applied the same filters, they increased the number of identified conserved SNP-CNEs to 558 [36].

Phenotypic screening of genetically diverse species can reveal CNEs with specific trait associations. Such associations allow us to identify CNEs altered in human diseases, for example, the disruption of an *IRF6* enhancer-binding site in cleft lip and palate [64] and a mutation in a *RET* enhancer in Hirschsprung disease [65]. Non-coding mutations can also cause nuanced changes in gene expression that may contribute to the spectrum of disease severity.

GWAS-derived SNPs embedded specifically in gene regulatory regions conserved to zebrafish are further expected to lie next to the orthologous gene(s) in both human and zebrafish, based on synteny. To exemplify the relevance of syntenic regions, Madelaine et al. identified SNPs associated with human diseases, such as rs11190870, mapped in a CNE close to *LBX1*, previously identified as responsible for scoliosis [36]. Madelaine et al. also questioned whether zebrafish could phenocopy aspects of human defects caused by the dysregulation of CNEs. To answer this, they applied CRISPR-Cas9 mutagenesis to delete an identified CNE previously associated with retinal blood vessel formation. Upon deletion of this specific CNE, zebrafish larvae remarkably showed disrupted blood vessel networks, vasculature defects in the retina, and downregulation of *microRNA-9*, rather than *MEF2C* as predicted by the original GWAS. Under *miR-9* depletion, the zebrafish retinal vasculature formation was compromised [36]. This work demonstrated how the analysis of conserved sequences in zebrafish can reveal the biological function of non-coding GWAS SNPs, which represents a major challenge for biomedical research.

Taking advantage of synteny to predict causality, Clément et al. extensively mapped CNEs on the entire genome-scale and predicted enhancer-gene interactions using a method previously applied to only the human chromosome X, called PEGASUS [66]. PEGASUS identifies all protein-coding genes in a 1 Mb radius around CNEs, then computes a linkage score for each gene, reflecting the evolutionary conservation of synteny between a CNE and a particular gene [67]. PEGASUS can map enhancers located in intronic regions of a different neighbor gene, and its predictions agree with those of genome-wide in vitro assays (Hi-C, Chip-Seq) in up to 42% of the cases. CNEs identified from Clément et al.’s work differs from those of Madelaine et al.’s due to the level of conservation of CNEs (the former compared 35 vertebrates for the identification of human CNEs, and six species for zebrafish CNEs) and the identification of orthologous regions. Because of the evolutionary distance between humans and zebrafish, some orthologous regions are difficult to align and impossible to detect. The authors used the spotted gar genome to identify additional orthologous CNEs between humans and zebrafish [66]. Although they had not aligned GWAS SNPs to identify those located in or proximal to CNEs, their syntenic genomic regions with or without full conservation with zebrafish could help in dissecting non-coding regions. It has been shown that despite low evolutionary conservation in non-coding genome sequences between humans and zebrafish, the last can still be used to test enhancer activity of putative human cis-regulatory elements [68,69], and therefore could be useful for testing relevant human CNEs.

A relevant recent work by Hirsch et al. from Birnbaum’s lab showed that structural variants (SVs) disrupting protein-coding sequences can also function as cis-regulatory elements [70]. They showed that craniosynostosis patients with SVs containing the Histone deacetylase 9 (*HDAC9*) protein-coding sequence are associated with disruption of *TWIST1* regulatory elements that reside within the *HDAC9* sequence. Based on SVs within the *HDAC9-TWIST1* locus, they defined the 3′ HDAC9 sequence (~500 Kb) as a critical *TWIST1* regulatory region, encompassing craniofacial *TWIST1* enhancers and CCCTC-binding factor (CTCF) sites. Deletions of either *Twist1* enhancers (eTw5–7Δ/Δ) or CTCF site (CtcfΔ/Δ) within the *Hdac9* protein-coding sequence in mice led to decreased *Twist1* expression and altered anterior/posterior limb expression patterns of Shh pathway genes. The authors found that *Twist1* frequently interacts with several regions encompassing enhancer candidates. Then, they tested these candidates for functional activity using zebrafish enhancer transgenic assay, in the craniofacial tissues of zebrafish embryos. The zebrafish enhancer assay showed that some candidate enhancers drove specific GFP expression in branchial arches and other facial bones. By this, Hirsch et al. provided an insight into the spatiotemporal regulatory network that controls *Twist1* expression in the developing craniofacial tissues [70]. These studies exemplify the promise of functional studies in animal models for deciphering non-coding regulatory regions (by deleting or disrupting them) and understanding disease causality, which remains a major challenge as of now.

### 4.3. Animal Modeling

One of the goals of in silico and in vitro methods is to narrow down candidates and translate findings to in vivo models. Animal models have contributed to our understanding of mechanisms of human diseases, such as how the genetic analysis of diversity in outbred mice facilitates the identification of novel loci in humans [71]. However, the value of animal models in predicting the effectiveness of drug treatments in the clinic has remained controversial. Animal models, including genetically modified ones and experimentally induced pathologies, might not accurately reflect disease in humans, and therefore might not predict with sufficient certainty what will happen in humans. For example, thalidomide, which was developed in the 1950s and originally used as a sedative, was soon being prescribed to expectant mothers for treating morning sickness, and it caused developmental anomalies (shortened, absent, or extra limbs and other defects) in unborn children. When tested in mice, however, thalidomide did not cause any defects [72]. However, animals are still heavily used to investigate human diseases and predict human responses to chemicals and drugs.

Most recently, Swan et al. explored data from 3823 mutant mouse strains for BMD [73]. A total of 200 genes were found to significantly affect BMD, including 141 genes with previously unknown functions in bone biology. In total, 19 of these 141 genes also caused skeletal abnormalities. Examination of the mouse BMD genes underscored OP-relevant pathways, including vesicle transport, and together with in silico bone turnover studies and the in vivo validation in osteoclasts and osteoblasts, resulted in the prioritization of candidate genes for further investigation [73].

Hyperglycemic zebrafish models are used as well in diabetes-related studies [74]. Hyperglycemic induction methods in zebrafish have been established; one of the most prominent knockout models of hyperglycemia is glyoxalase1 (*glo1-/-*) fish. Thus, validated diabetic induction methods are suggested to help researchers confirm their proposed models [74].

To date, even with the advanced Mouse KO and phenotyping collaborations, such as the International Mouse Phenotyping Consortium [75] and the Origins of Bone and Cartilage Disease consortium [76], only a minority of these genes were tested for relevance to increased bone fragility. Reaching such an objective requires a systematic prioritization and standardized phenotyping, which in turn asks for additional collaborative efforts, as outlined in the recent efforts from GEMSTONE Consortium [40,77]. Furthermore, with a large number of associated BMD loci and number of genes to be functionally validated, along with the labor, time and high costs for generation of new mice knockouts, mice are impractical models for parallel functional throughput evaluation of candidate genes. Small teleosts bring solutions to circumvent these limitations.

To complement the usability of fish to model skeletal diseases, a recent review summarized the rapid emergence of zebrafish models of rare skeletal diseases during the last 10 years [78]. Many available genetic models have been summarized [61,79,80], and phenotyping methods have been described for modeling bone disease using fish [81].

## 5. Zebrafish as Animal Models to Accelerate Discoveries of Human Skeletal Conditions

Zebrafish have emerged as model organisms in bone research within recent decades, being choice models to confirm gene discovery and to perform functional analysis of human variants, often providing viable alternatives to mouse lethal alleles. One of the most fascinating advantages of zebrafish is their transparency (Figure 3A). Zebrafish are transparent during development, permitting cell trackability by using transgenic reporter lines labeling specific cell types (i.e., osteoblasts, osteoclasts, and chondrocytes) [40]. Analysis of bone and cartilage features can be initiated by as early as three days post-fertilization (dpf), also allowing drug screenings for bone and cartilage formation and modifiers [82]. The large number of eggs that each zebrafish pair produces weekly (~200 eggs), similar physiology, genetics, and shared drug responses with humans highlight the advantages of zebrafish as emergent models in drug discoveries [83]. While in vivo analysis of the behavior of key bone cells during bone remodeling in mice is invasive, by necessity depending on surgery and intravital imaging techniques [84], in fish osteoblasts and osteoclasts can be easily and non-invasively monitored in vivo using cell-specific reporter lines [85] (Figure 3A). Fish models are notable for understanding the craniofacial genetics [86,87], skeletal diseases, and even for the co-occurrence of craniofacial and limb disorders [88]. Contrary to mice studies, the development of the craniofacial skeleton, including the skull, can be observed in vivo and non-invasively, allowing researchers to investigate cellular behavior changes underpinning common craniofacial developmental conditions [89].

In 2016, Kague et al. took advantage of the transparency offered by zebrafish to show the complete development of the cranial sutures in vivo for the first time [89]. The same work demonstrated ectopic suture formation and the development of Wormian bones (bones surrounded by sutures) in zebrafish lacking the transcription factor *sp7*. Remarkably, these findings in zebrafish precisely recapitulated those observed in a reported patient affected by osteogenesis imperfecta recessive that carried a homozygous loss-of-function mutation in *SP7* [89]. Therefore, zebrafish have efficiently modeled craniosynostosis, a premature fusion of the cranial sutures. These models are indicated in Table 2. Furthermore, zebrafish were recently used to validate new BMD candidate genes identified from the recent skull-BMD-GWAS, providing functional evidence that skull-BMD loci provide a repertoire of genes to be tested for involvement in craniofacial malformation and BMD regulation. Medina-Gomez et al. performed the largest skull-BMD-GWAS meta-analysis identifying 59 loci, from which 4 loci were novel [7]. The group tested *zic1*, *atp6v1c1a*, *atp6v1c1b*, *prkar1a,* and *prkar1b* in zebrafish, unprecedently showing that loss-of-function of each of these genes could lead to craniofacial malformation and abnormal BMD, thus, exemplifying the use of zebrafish to elucidate genes playing a role also in developmental craniofacial conditions.

With the transparency gradually lost during development, the adult zebrafish endoskeleton becomes covered by musculature and mineralized scales. While in vivo imaging can still be used to study bony elements located superficially and permanently accessible in fish (i.e., skull, fins, opercula, jaw, and scales) (Figure 3A), to analyze the endoskeleton other technologies, similar to those used in mice, are applied. For example, Alizarin Red staining is used to visualize mineralized bones, and microCT (uCT) allows 3D morphological analysis and precise BMD measurements (Figure 3B,B’,C,C’).

Zebrafish passed through an additional whole genome duplication. Zebrafish gene duplicates (orthologs to mouse/human genes) may functionally balance each other, so single knockouts might be viable at least during embryonic and early larval periods, allowing basic analysis of skeletal development. Zebrafish orthologs sometimes manifest sub-functionalization, so that viable loss of function alleles of a specific paralog can be obtained and studied [61,81]. Each model is not without its limitations. Indeed, it is not known whether zebrafish show similar characteristics of bone remodeling as in humans, which in the bone marrow cavity of humans a canopy is formed by the bone lining cells to isolate the bone remodeling multicellular unit. Zebrafish lack long bones and bone marrow, their bones are hollow, filled by fat cells (in the vertebral column, by notochordal cells). Whether canopies are formed in these bone areas, will need to be investigated. Chondrocyte differentiation modeling growth plates is found in the jaw bones and hypurals (adjacent to the caudal fin) [130,134,135]. Another limitation of zebrafish is the absence of the typical trabecular bone observed in mammals. Given miniaturization (very small structures of small species), trabecular bones are reduced to only a few trabeculations protruding from and surrounding the vertebral centra of zebrafish spines [136]. Their specific development in fish have not yet been characterized. However, they develop after the complete formation of the vertebral column centra and accompany the growth of the fish during life. In larger species, such as in salmon, more trabeculations are observed surrounding the vertebral centra (Figure 4). A reduction in trabeculation in fish, compared to humans, is attributed to the lower mechanical load of aquatic species in comparison with that of terrestrial. It is important to understand the differences between trabecular vs. cortical compartmentalization in fish and humans, since anti-osteoporotic drugs tend to act more at one compartment depending on their mechanism of action.

### 5.1. Genetic Engineering in Zebrafish to Test Gene Causality

Although the evolutionary distance between zebrafish and humans date back to 450 million years, about 71% of zebrafish genes are conserved with humans. Moreover, when considering only genes involved in diseases, zebrafish have ~80% of disease-causing genes conserved with humans. For this reason, zebrafish have been an attractive model in forward genetic screening using Ethyl-N-nitrosourea (ENU)-driven chemical mutagenesis. In addition, in reverse genetics by targeting specific genes using gene-editing techniques such as zinc-finger nucleases (ZFINs) [138,139,140], TALEN [141] and more recently, CRISPR/Cas9 [142]. In the early 2000s, in the absence of tools for efficient targeted mutagenesis, morpholinos (antisense oligomers) became a popular tool for gene knockdown, often able to phenocopy well-characterized zebrafish mutants. However, morpholinos only offer a transient knockdown of genes of interest (up to 4 days post-fertilization) and very frequently off-target effects resuming worse phenotypes than mutants [143]. The CRISPR/Cas9 genome editing technology revolutionized how genes are functionally tested in zebrafish, adding speed, reproducibility, and high efficiency, which allow us to phenocopy knockout phenotype already in the first generation, so called crispants.

We and others demonstrated that G0 knockouts reliably recapitulate complex mutant phenotypes, such as osteopenia and osteogenesis imperfecta [93,144]. Most recently, an effective G0 knockout method for rapid screening of behavior and other complex phenotypes was proposed. To facilitate rapid genetic screening, Kroll et al. developed a simple sequencing-free tool to validate gRNAs and a highly effective CRISPR-Cas9 method capable of converting >90% of injected embryos directly into G0 biallelic knockouts [145]. This G0 knockout method cuts the experimental time from gene to behavioral phenotype in zebrafish from months to one week [145]. Crispants have been compared to knockout mutants for genes involved in osteogenesis imperfecta (*PLOD2*) and osteoporosis (*LRP5*), and were able to reproduce similar phenotypic characteristics, such as low BMD, as those of mutants [93,144]. Zebrafish crispants represent powerful tools for rapid functional annotation of GWAS identified variants and to bridge the gap between association and causality. Once a knockout is generated, the deep phenotyping of zebrafish models is studied by different methods at the anatomical, cellular, and molecular levels and compared with human symptoms to clarify its potential utility.

### 5.2. Swimming from Bench to Bedside: Druggable Gene Targets

The identification of genes involved in skeletal diseases can result in novel treatments for osteoporosis, as was the case for denosumab, an anti-RANKL monoclonal antibody [146,147] and romosozumab, an antibody against the Wnt-inhibitor sclerostin, SOST [148]. And in fact, it is estimated that the success rate during the development of a drug is doubled with drug targets that have human genetic supports [149]. Meaning that GWASs offer an arsenal to boost drug development towards OP treatments, and towards effective anabolic treatments. However, none of the GWAS performed to date have led to new treatments for osteoporosis. This might be attributed to challenges to identifying causal genes for complex diseases, such as OP.

Interestingly, the first small-molecule inhibitor of BMP signaling, dorsomorphin, was one of the first compounds identified from a zebrafish chemical screening aiming to identify compounds that phenotypically reproduce loss of BMP signaling pathway (*alk8-/-*) [150]. Currently, dorsomorphin derivatives (ALK2 inhibitors) are on a clinical trial for the treatment of heterotopic ossifications in fibrodysplasia ossificans progressiva. Drugs identified in zebrafish have also the potential to go straight to clinical trials: the first “aquarium-to-bedside” example of drug identified in zebrafish and tested directly in humans is *clemizole* to treat patients with Dravet syndrome. Dravet syndrome is a catastrophic childhood epilepsy with early onset of seizures, caused primarily by mutations in *SCN1A*. Zebrafish *scn1* mutants recapitulated Dravet syndrome and were thus used in a phenotypic screening resulting in the identification of Clemizole. Clemizole binds to serotonin receptors; its antiepileptic activity can be mimicked by other drugs acting on the serotonin signaling pathway (e.g., *lorcaserin*). Griffin et al. tested *lorcaserin* in Dravet patients and showed reductions in seizure frequency and/or severity, similar to Clemizole in the fish [151]. These two examples demonstrate the power of drug screenings that zebrafish bring for different human conditions. Over ten compounds identified from zebrafish phenotypic drug screenings are currently in clinical trials (reviewed by Patton et al. [83]). Besides speeding up the identification of causal genes using genetic tools, zebrafish can be used as tools to test osteoporotic candidate drugs and help to evaluate drugs against bone fragility symptoms. Such assays are conducted by the addition of chemical reagents to system water or embryo medium and a subsequent larval skeletal phenotypic study [78,152]. Likewise, soluble drugs can be applied in the water of adult zebrafish followed by adult phenotypic assessment [153].

Interestingly, for some drugs, zebrafish are able to recapitulate the effects observed in humans better than mouse models do. This is the case for thalidomide, mentioned above, which safety was originally tested in mice, but when administered to pregnant mice, the drug does not reach teratogenic levels in embryos, thus not causing birth defects as in human [72]. It was only years later, using zebrafish and chicks, that researchers identified the teratogenicity of thalidomide on limb malformation, as zebrafish precisely recapitulated limb defects observed in children [154]. If initial tests involving thalidomide were performed in parallel in zebrafish, its teratogenic effect in children could have been identified earlier.

### 5.3. Fish-Specific Environment

The zebrafish models of damaging environmental exposures exist for tens of years, most prominently for toxicology research. Specifically for osteoporosis, several non-genetic models were established by the iron-stress [155], by prednisolone treatment [156], and through movement restriction experiments [157]. Fish models of induced OP have been recently reviewed [158]. It is worth mentioning that zebrafish have been increasingly used to test potential natural compounds for the treatment of osteoporosis. Because small fish offer a rapid and non-invasive in vivo platform for bone formation, the model has been used to screen a variety of natural compounds derived from traditional herbal medicine, especially Chinese medicine, and to validate in vitro findings. Natural compounds have been tested using glucocorticoid, glucose-induced OP and *rankl*-induced OP (medaka) to validate their anti-resorptive and anabolic activities [158]. For example, liquiritigenin (a flavonoid extracted from *Glycyrrhiza glabra* roots) has a positive effect on osteoblast differentiation, as well as halts osteoclast activity in vitro. Carnovali et al. tested liquiritigenin in zebrafish, revealing increased bone formation rates in a dose-dependent manner and reduced TRAP staining in zebrafish scales of glucocorticoid-induced OP [159]. A similar approach was used to analyze natural foods with antioxidant properties (curcuma, papain, bromelain and black pepper), which showed anabolic effects during development and reduced osteoclastic activity in zebrafish scales [160]. Zebrafish have been of interest to also test marine alkaloids [161], which altogether shall boost the discovery of potential alternative therapeutics.

Although the zebrafish skeleton has a reduced requirement for resisting mechanical load due to residing within an aquatic environment, several studies have demonstrated that swim training can influence the timing of skeletogenesis in zebrafish larvae [162], as well as increase bone formation and mineralization in the adult vertebral centra [163]. Adaptation of the zebrafish skeleton to mechanical forces is reminiscent of that in mammalian models of bone adaptation to exercise, spinal cord injury, microgravity, or other modes of mechanical loading/unloading [157,164]. Although gravity is lower in the water than on the surface, fish still swim against viscous water, therefore effects of this complex hydrodynamic system can be seen on the cranial shape and vertebral curvature [163]. Lower gravity in fish is consequently reflected by different bone mineral density and rare fracture events associated with the vertebral column [165].

Also, an important work by Witten et al. reminds us about the commonalities and specifics of small teleost biology in response to their environments [166]. Zebrafish and medaka belong to two different orders within the ray-finned fishes and have a long evolutionary divergence time [166]. The teleost skeleton is a prominent place for lipid storage [166]. A relatively limiting factor is the fact that teleost and other primary aquatic gnathostomes (vertebrates with jaws) are able to effectively obtain calcium from the water via the gills but they depend on dietary phosphorous intake for the mineralization of the skeleton [166].

### 5.4. Osteoporosis Phenotypes to Study in Zebrafish: A Work in Progress

Osteoporosis is characterized by an imbalance between bone formation and bone resorption, where osteoclast activity exceeds that of osteoblasts. The behavior and interaction of bone cells that are commonly studied using cell cultures, can be easily studied in fish because they are see-through during a relatively long period of their development. Larvae zebrafish are often used to analyze osteoblast behavior, bone matrix deposition and mineralization [91]. In mammals, osteoclast maturation involves multinucleation at a late phase of osteoclast differentiation. However, those osteoclasts that remain mononucleated keep their characteristic phenotype, with expression of osteoclast-related markers such as TRAP and Cathepsin K, and retain low levels of bone resorption ability [167]. As in mammals, both mono- and multinucleated osteoclasts in zebrafish contribute to allometric bone growth and express osteoclast markers, such as TRAP [168]. Zebrafish mononucleated osteoclasts are present from the second week post-fertilization, they are predominant in juvenile stages, and at thin skeletal elements (neural arches, nasal bones) of adults. Multinucleated osteoclasts are observed at around 40 dpf and are the predominant osteoclast types of adults [168]. Medaka has provided proof of principle showing concomitant osteoblast/osteoclast imaging in a RANKL overexpression model induced by heat-shock. Upon heat-shock, osteolysis is induced and bone remodeling can be followed in vivo [85]. Such a model allows the validation of new osteoporosis targets and the identification of novel molecular players [94].

Here it should be noted about the zebrafish exoskeleton (scales and fins) for the needs of comparative anatomical understanding. As part of the dermal skeleton, these structures harbor a mineralized matrix and serve as sources for rapid evaluation of bone formation and mineral deposits [169,170]. In addition, with the ability to regenerate, scales and fins allow us to explore the molecular profile of osteoblasts during regeneration [171]. With the amenability to culture scales for a few days, these dermal bones have been of interest to test anabolic factors, and the system is able to achieve a relatively high-throughput fashion [172]. On the other hand, the small and first developed ossicles of zebrafish are another alternative system to test anabolic effects. Larvae can be concomitantly used to test levels of drug toxicity and off-target effects (i.e., heart oedema, vascular or neurological system) [82,83]. Due to the similar physiology and drug metabolism, zebrafish poise as powerful model systems to test future pharmacological modifies with the potential to treat skeletal conditions.

Adult zebrafish models to study osteoporosis have only been developed recently, alongside the study of the ageing zebrafish spine [92,173,174]. The similarity between the techniques and interpretation of the fish, mouse, and human skeletal phenotyping was recently described [77], as well as the necessity to reach a consensus and prepare guidelines for standard measurement of the small-fish OP-related measurements. Recently, Kague et al. provided strong support of osteoporosis modeling in zebrafish, showing that aged zebrafish spines display increased susceptibility to fractures and have bone quality deterioration (tendency towards reduction of BMD, increased bone mineral heterogeneity and poor collagen organization) reminiscent of osteoporosis in mammalian models [92]. Kague et al. also demonstrated that bone deformities in the zebrafish vertebral column endplates during ageing can lead to falsely elevated BMD and underdiagnosis of osteoporosis, as observed in humans with degenerative changes of the lumbar spine [175]. However, under genetic manipulation, even young zebrafish provide consistent and compelling models for osteoporosis, with BMD changes easily detected by standard techniques. This has been demonstrated with several mutants, including zebrafish *sp7-/-* and *lrp5-/-* [92,93], etc. By using high-resolution 3D imaging (<0.1 μm, synchrotron radiation) one can also retrieve osteocyte lacunar profile [92,108]. Table 2 shows examples of genetic models for human skeletal diseases.

The endpoint of osteoporosis is the occurrence of a fracture. While the prediction of fractures in zebrafish is not comparable to that of humans, where FRAX (fracture assessment tool) is used; genetically modified zebrafish offer alternative ways to study fracture occurrence, prediction and fracture healing [77]. The zebrafish bony fin rays, known by fish experts as lepidotrichia, are under dynamic mechanical load during the fish swim, and are regions of interest to monitor bone fractures. Aged zebrafish show increased numbers of spontaneous fin fractures, detected by the presence of callus [95]. Calluses are also observed prematurely in young zebrafish carrying a loss-of-function mutation in *wnt16* or *bmp1a1* [95,176]. Moreover, a fracture can be induced by applying pressure on fish fins and fracture healing can be monitored longitudinally in vivo (Figure 3A). It has been shown that zebrafish display similar phases of fracture healing as in mammals, however, endochondral ossification is minimal [176]. Interestingly, *bmp1a* fractures are prone to non-union [176], serving as a model to test modulators of fracture non-union. Therefore, the study of osteoporosis in fish can be performed from the behavior of key bone cells, osteoblasts and osteoclasts, to bringing pharmacological opportunities to treat fracture non-union. In the endoskeleton, fractures are rarely detected in the vertebral column, but the ribs are susceptible to fractures, as frequently reported in mutants modeling osteogenesis imperfecta [89,107,108]. Therefore, the ribs are another anatomical region to quantify fractures numbers and recurrence, therefore, of relevance for osteoporosis studies.

The *WNT16* locus has been explored using zebrafish by several groups, serving as a good example of GWAS findings followed by functional validation in fish. Since the GWAS signal on 7q31 was discovered [13,177,178], it remained an open question about which gene was responsible for this signal, *WNT16*, *FAM3C*, or *CPED1* [179]. The original functional work was published using mouse models [179,180]. The authors demonstrated a role of *WNT16* in the regulation of cortical bone thickness through increased resorption and reduced periosteal bone formation [180,181]. In 2021, three papers based solely on zebrafish were published on *wnt16* loss-of-function, incrementing our current knowledge about the gene in bone and its pleiotropic effects. Firstly, McGowan et al. showed that caudal fins of *wnt16* zebrafish mutants were highly susceptible to fractures, with a 10 fold increase in numbers of calluses in mutants when compared to WT of a similar age [95]. The authors also induced fractures and followed fracture healing over 15 days. They demonstrated that despite normal bone healing, the recruitment of osteoblasts was compromised in mutants during the early stages of repair [95]. While McGowan et al. focused on bone fractures, Qu et al. studied the skeletal structure of a second *wnt16* mutant, reporting reduced bone density, curved spines and abnormal jaw [96]. The same group performed RNA-seq, identifying differentially expressed genes associated with mTOR, FoxO and VEGF pathways [96]. Finally, recent work by Watson et al. interrogated the biology behind the pleiotropy at the locus (which was associated with BMD and lean mass) [97]. The authors detected reduced vertebral centra volume and length, and the fish were shorter. Through single-cell RNA-seq the authors could cluster cells belonging to muscle sub-compartments, reporting differential expression of *wnt16* in dermomyotome and sclerotome. By performing cell tracing experiments, they demonstrated that somatic wnt16+ cells are muscle precursors that contribute to muscle fibers. Lastly, they performed an analysis of lean mass in *wnt16* mutants, which exhibited an altered distribution of lean mass and altered myomere morphology. This study suggests that *wnt16* exerts a pleiotropic effect on bone and lean mass. Together, all three *wnt16* zebrafish studies are complementary in concluding that *wnt16* is necessary for bone mass and/or morphology, serving as a showcase for zebrafish functional studies to interpret findings from GWAS.

## 6. Conclusions and Future Directions

When it comes to the functional validation of bone-disease-related candidate genes highlighted through different venues, including WGS and GWAS, small teleost fish provide an unprecedented speed and precision not achievable with mammalian in vivo systems. Zebrafish are able to recapitulate human bone diseases and offer advantages for functional validation and deep phenotypic studies of osteoporosis-related genomic findings to an extent which they are preferable models, in many of the cases. It is expected that studies incorporating the fresh-water teleost model will continue increasing. We highlight key areas where we foresee the use of zebrafish in the osteoporosis research field.

The in vivo aspect of how cells behave under normal conditions, or in diseases and after pharmacological interventions is an important area of research that will optimize future osteoporosis treatments [84]. This has been neglected given the technological difficulties of live imaging of bone resorbing activity in rodents. Zebrafish offer a whole 3D in vivo system where not only osteoblasts and osteoclasts can be monitored longitudinally, but also other tissues, such as muscle, vasculature, immune system’s cells and innervation to these areas. Zebrafish have the power to extend our knowledge about cell interaction, the dynamics of bone formation and maintenance under different genetic and/or pharmacological conditions. While the RANKL overexpression system has been implemented in Medaka to induce osteolysis and bone remodeling [85], the same system could be easily implemented in zebrafish to follow in vivo bone remodeling. As noted, several drawbacks have to be overcome. Zebrafish bones are hollow; they lack bone marrow. It is unclear whether the bone remodeling multicellular unit works as in humans, whether it presents a “canopy” formed by the bone lining cells. In addition, bone trabeculation and possible phenotypes in zebrafish should be explored, in parallel to the molecular mechanisms of trabecular formation. Contrary to osteoblast studies, those involving osteocytes are still very limited in fish models, due to the challenges of image acquisition (necessity of higher resolution imaging, such as Synchrotron) and the lack of osteocyte reporter lines for in vivo imaging attempts. Osteocyte studies in zebrafish could also benefit from the recent advancement in the characterization of osteocyte transcriptome profiling from mammalian models [182], that have highlighted important genes that could be used to establish reporter lines targeted toward studies in fish. Kague et al. used artificial intelligence to automate 3D analysis of osteocyte lacunar profile, therefore handling computational tools to the community to further facilitate osteocytes studies in zebrafish. Increasing the availability of tools to study osteocytes in zebrafish will allow us to explore in detail the effect of bone conditions on osteocyte behavior and function.

It is worth mentioning the use of zebrafish exoskeleton and fins for supplementary skeletal studies. With the amenability of scales to be cultured in vitro and the regenerative capacity of fins, these dermal bones have been of interest to test osteoblast anabolic factors and are capable of relatively high-throughput experiments [153]. Together with the ability of larvae to be used to test drug toxicity levels and off-target effects, these systems offer efficient screening platforms to test new bone anabolic compounds.

Despite a few studies performed in the ageing zebrafish skeleton [92,173,174], many questions remain unanswered, and these include the dissection of the molecular signaling pathways and key genes involved in ageing of the musculoskeletal system as well as their overlap with human tissues compromised by degenerative diseases. By analyzing to what extent ageing alters the fish skeletal regenerative capacity, one could also highlight potential molecular players in degenerative conditions. As ageing studies in fish continue to emerge, it will be worthwhile to answer whether ageing recapitulates signaling pathways of early development through comparative transcriptomic analysis and additional functional work. One limitation of ageing studies is the waiting time until the fish is considered old (~2.5 years for zebrafish). Therefore, whether genetic manipulation of key genes could induce premature skeletal ageing would accelerate such studies. Premature ageing has been reported in *klotho* zebrafish mutants [183], telomerase-deficient zebrafish [184] and progeroid zebrafish models [185] however premature skeletal degenerative changes have not yet been studied in these mutants. Notably, the normally aged zebrafish (45 months old) exhibited ectopic ossification inside or outside of vertebrae [92], also reminiscent of the aged human spines.

Given the high number of loci identified to play a role in osteoporosis, we consider this specific gap in knowledge one of the top beneficiaries of what zebrafish can currently offer. The high efficiency of genomic editing in zebrafish using the CRISPR/Cas9 system, as mentioned above, allows us to accelerate our genetic validations using the first zebrafish generation (crispants) that are able to phenotypically recapitulate knockouts. This would allow us to implement the closest to high-throughput genetic screenings that a living vertebrate can achieve (Figure 5). With the increased incorporation of Artificial Intelligence (machine learning) for imaging analysis, it is realistic to predict the combination of machine learning with zebrafish crispants to deliver rapid platforms to validate genes associated with BMD and osteoporosis as well as other complex bone or cartilage conditions. Remarkably, with such tools, it will be possible to circumvent some of the challenges of GWAS SNPs mapped either in large non-coding regions or those in gene-dense regions. Zebrafish will allow us to test several neighbor genes within the same locus, fetching the power to interpret gene causality. Because the CRISPR/Cas9 system can also be designed to target multiple genes at the same time, another approach that can be used is the implementation of multiplex crispants to analyze additive phenotypes of polygenic loci. Therefore, as far as genetic conservation goes between zebrafish and humans (~70% genes conserved), zebrafish are able to add functional annotations for most of the genes associated with BMD. Currently, zebrafish knockout models prevail over knock-ins for efficient analysis of GWAS candidate genes. An increasing number of studies have demonstrated the feasibility of gene knock-in in zebrafish [186,187,188], however, mutagenesis efficiency is not yet comparable with those of knockouts. The low efficiency of knock-ins compromises their feasibility for high-throughput studies. Additionally, the effect size of knock-ins may be smaller, reducing the power to analyze gene function. Improvements regarding the engineering of gain-of-function zebrafish lines would help to validate gene function. The skeletal field will also benefit from the development of emergent technologies towards cell-specific conditional zebrafish knockouts using Cre/LoxP or CRISPR/Cas9 systems [189,190].

Finally, non-coding variants regardless of their conservation within fish clades can be tested using zebrafish [69,192]. It has been shown that mosaic transgenic analysis (G0s, first-generation) can be also used to test enhancer activity [192]. As for crispants, the analysis of CNEs in mosaic zebrafish G0s also has the potential to accelerate the functional annotation of enhancer elements corresponding to GWAS SNPs [193]. Testing enhancer activity and comparing with the impact of single nucleotide changes within these sequences are already possible in zebrafish. While most of the designed engineering approaches to test enhancers in zebrafish rely on variable genomic site integration, recent improvements in the technology permit to control genomic integration, reducing noise and increasing reproducibility [194]. Given the transparency of zebrafish, the activity of enhancer elements can be monitored in a temporospatial way, at least during development. Larger deletions of highly conserved non-coding sequences can be performed through CRISPR/Cas9, however, there is limited availability of target sites recognized by the Cas9. To evade this challenge, the zebrafish field has been showing improvements in efficiently testing other types of Cas proteins (Cas12a/Cpf1, etc.) [195,196], therefore increasing target sites within non-coding regions for efficient CRISPR editing, and base editing for efficient knock-ins. Assaying enhancer activity compared with the expression pattern of neighbor genes, alongside in vitro machine learning, will assist in mapping enhancers to their target genes during the early development of the musculoskeletal system, and beyond.

We have mentioned a few areas where zebrafish are being implemented to add power and speed to the functional evaluation of osteoporosis-associated genes reinforcing them as a choice model organism to support bone-GWAS and help to bridge the current gap between association and causality. Having said this, it is notable that the legislators and the anti-animal-research activists are frustrated that, despite a legal requirement to use methods that do not involve animals, their development and uptake remains slow. Recently, the EU Parliament urged its constituents to accelerate the transition to a research system that does not use animals, pushing for alternative testing methods and phasing out the use of animals in research and testing. This is in line with the previous decade’s conventions and efforts focused on reducing, refining and replacing (3R) procedures on live animals for scientific purposes, as soon as it is scientifically justifiable and “without lowering the level of protection for human health and the environment”. It is advised that the scientists must be trained in using advanced non-animal models and in sharing best practices regarding the 3Rs. Thus, for example, both positive and negative indicators of animal welfare must be reported. As such, there is a pressing need for studies of welfare to more fully assess negative and positive factors to truly capture an animal’s experience of change in their health environment. Both the future of small-fish modeling in general and osteoporosis genetic research would benefit from this knowledge.

## Figures and Tables

**Figure 1 genes-13-00279-f001:**
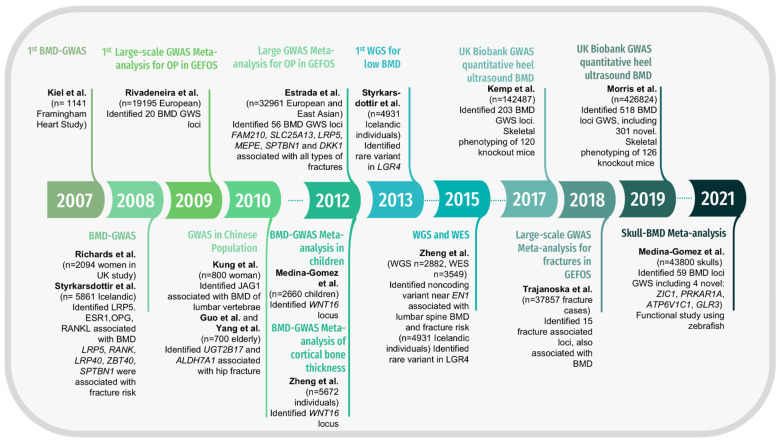
Timeline highlighting important milestones of GWAS for BMD. GWS (Genome-wide significant).

**Figure 2 genes-13-00279-f002:**
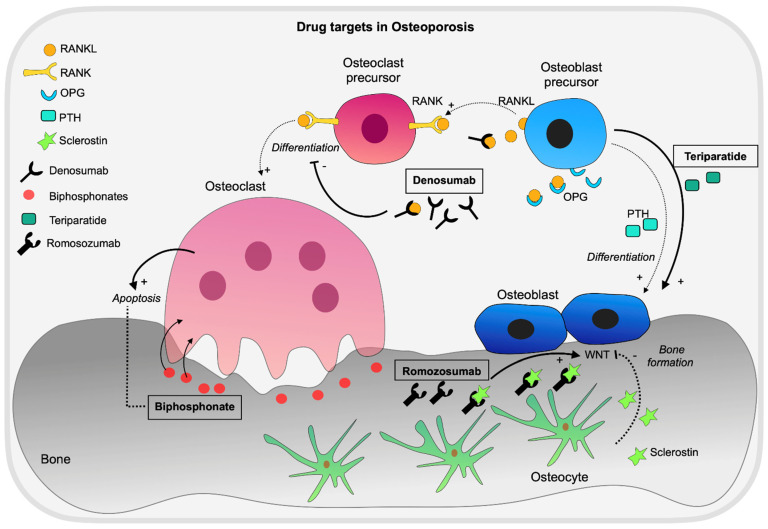
Overview of drug targets and current therapeutics for osteoporosis. Denosumab, Romosozumab, Teriparatide, and Biphosphonates are the most used drugs for OP, highlighted with boxes and in bold. Denosumab inhibits the differentiation of osteoclast precursors to mature osteoclasts and the function of mature osteoclasts. Romosozumab is the most effective treatment for OP, it is an anti-sclerostin antibody that allows increased bone formation rates and reduced osteoclastic activity. Teriparatide is an anabolic agent, homologous to parathyroid hormone, positively regulating osteoblast differentiation and bone formation. Bisphosphonates are the most prescribed therapeutics, they bind to hydroxyapatite of bony surfaces and are absorbed by osteoclasts during resorption, leading to impairment of osteoclast activity and apoptosis.

**Figure 3 genes-13-00279-f003:**
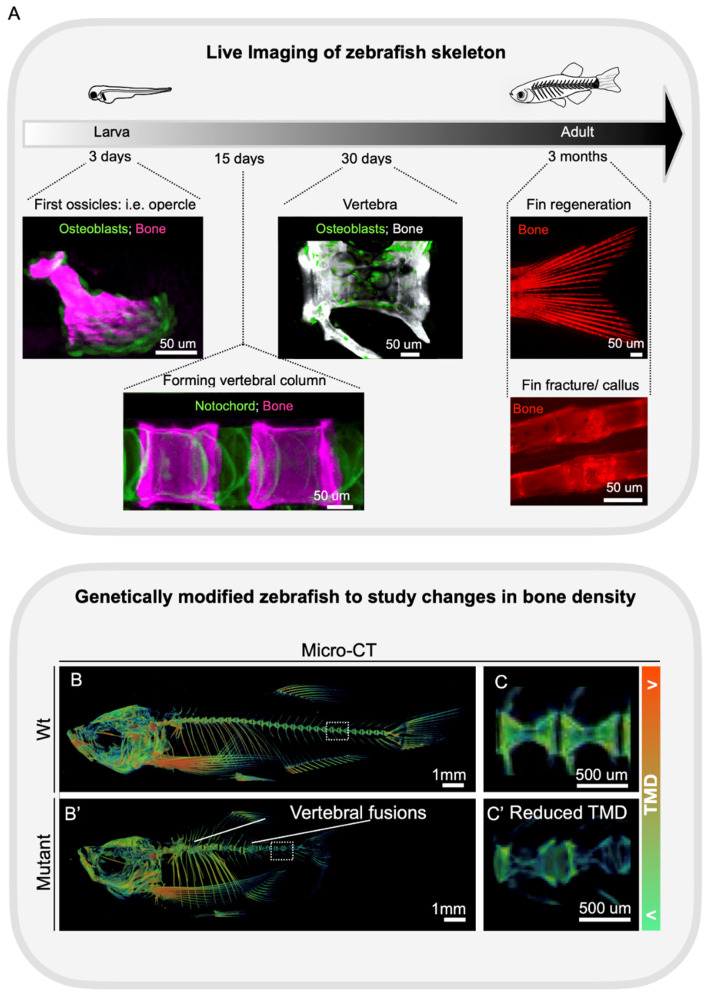
Phenotypic skeletal analysis in zebrafish: from in vivo cell behavior to skeletal maintenance. (**A**) The transparency of zebrafish allows us to visualize bone cells in vivo during skeletal development. Longitudinal in vivo studies frequently involve caudal fin regeneration upon amputation and fracture healing in adults. Examples include live images of reporter lines labeling osteoblasts, live staining with Alizarin Red to label the bones. (**B**) Images of volume rendering from micro-computed tomography (uCT) of adult WT and mutant (chordoma model [90]), as an example of skeletal abnormalities in mutants (vertebral fusion and lower TMD). Variations of TMD are shown color-coded. TMD scale representing <(lower TMD values) and >(higher TMD values). Higher magnification of the dashed box region in the spine is shown in (**C**). Wt (**B**,**C**), mutant (**B’**,**C’**). Scale bars are as indicated.

**Figure 4 genes-13-00279-f004:**
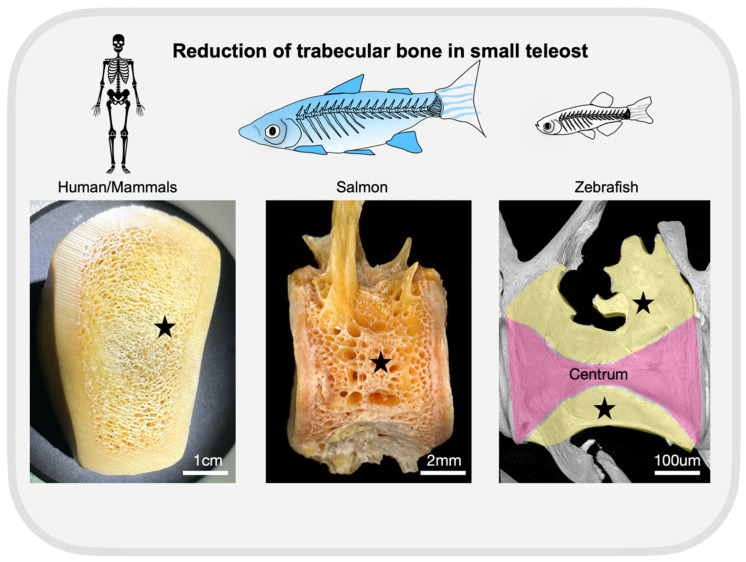
Trabecular bones in fish are located surrounding the centrum while in mammals they are surrounded by cortical bones. Small teleost species show only a few trabeculations (stars) when compared to larger fish. As an example of trabecular bone in mammals, we are showing a section of tibia from a dog, provided by the Histology Facility at the University of Bristol. The picture of salmon was adapted from [137] with permission from the correspondent authors. Note the reduction of trabeculations in zebrafish (synchrotron image, trabeculation shown in yellow and the centrum in pink). Scale bars are as annotated.

**Figure 5 genes-13-00279-f005:**
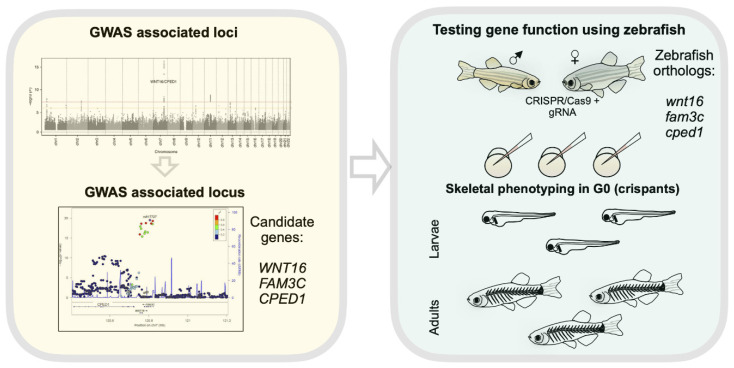
Zebrafish offer rapid functional evaluation of osteoporosis candidate genes identified through GWAS. On the yellow box (left side), we show a Manhattan plot (adapted from Medina-Gomez et al. [191]), followed by the 7q31 locus with genome-wide significant SNPs. Neighbors of *WNT16* (*CPED1* and *FAM3C*) are also candidates. As shown on the right side (green box), zebrafish have only one copy of each *wnt16*, *cped1* and *fam3c*. Zebrafish eggs can be injected with CRISP/Cas9 cocktail targeting each of the zebrafish orthologs. Skeletal phenotyping usually starts three days after injections, in G0s, and can be continued into adulthood.

**Table 1 genes-13-00279-t001:** WGS studies in bone density/osteoporosis.

Author/Ref	Study Sample	Age, Sex	Variant, MAF, Consequence	Effect Size	Gene (s)	Note
Shen et al., 2013 [21]	44 unrelated Caucasian adults	22 males and 22 females	NR	--	--	Did not replicate
Styrkarsdottir et al., 2013 [19]	Low BMD = 4931; Control = 69,034, from Iceland	NR	c.376C > T, 0.14–0.18%, nonsense	OR = 4.30 (low BMD)	*LGR4*	Could not replicate
Styrkarsdottir et al., 2016 [20]	Low BMD = 2894; Control = 206,875, from Iceland	NR	p.Gly496Ala, 0.105%, missense; p.Gly703Ser (0.050%), missense	OR = 4.61 (low BMD) OR = 9.34 (low BMD)	*COL1A2*	Could not replicate
Zheng et al., 2015 [14]	*n* = 2882 with. WGS; *n* = 3549 with WES; European	range of ages; male and female	rs11692564(T), 1.6%; non-coding	+0.20 S.D. (spine BMD)	*EN1*	OR = 0.85 for OP fractures (n cases = 98,742; n controls = 409,511)
Younes et al., 2021 [3]	3000 Qatari, Qatar Biobank	18–70 years old; 1442 males, 1558 females	Several; MAF ≥ 1%; mostly intronic	Low	*MALAT1*/*TALAM1*; *FASLG*; *SAG*; *LSAMP*; *FAM189A2*	Did not replicate

NR—none reported; blank—not available.

**Table 2 genes-13-00279-t002:** Fish genetic models for human skeletal diseases.

Human Disease	Fish Genetic Models	References
Osteoporosis	*atp6V1H*	[91]
	*sp7/osterix*	[89,92]
	*lrp5*	[93]
	*cxcl9l* *; *cxcr3.2* *	[94]
	*Rankl* overexpression *	[85]
	*Wnt16*	[95,96,97]
	*Rmrp*	[98]
	*galnt3*	[99]
	*mmp14*	[100]
	*meox1*	[101]
	*lrp4*	[102]
	*mafbb*	[103]
	*copb2*	[104]
	*megf6a*, *megf6b*	[105]
Osteopetrosis	*Pu.1* and *fms*	[106]
Osteogenesis imperfecta (OI)	*col1a1a* (chihuahua)	[107,108]
	*col1a2*	[109]
	*bmp1a* (frilly fins)	[110]
	*plod2*	[111]
	*sp7/osterix*	[89]
	*pls3*	[112]
Craniosynostosis and ectopic sutures	*cyp26b1* (dolphin and stocksteif)	[113]
	*tcf12* and *twist1*	[114]
	*fgfr3*	[115]
	*sp7/osterix*	[89]
	*zic1*, *atp6v1c1a*, *atp6v1c1b*	[7]
Fibrodysplasia Ossificans Progressiva	*acvr1/alk2*	[116]
Scoliosis	*cc2d2a*	[117]
	*kif6*	[118]
	*c21orf59*, *ccdc40*, *cctc151*, *dyx1c1* and *ptk7*	[119,120]
	*col8a1a*	[121]
	*ptk7*	[122]
	*dstyk*	[123]
	*cfap298*	[124]
	*sspo*	[125]
Osteoarthritis	*col11a2*	[126]
	*prg4*	[127]
	*gdf5*	[128]
	*nkx3.2*	[129,130]
Ectopic mineralization	*enpp1*	[131]
	*abcc6a*	[132]
Chordoma (bone cancer)	*HRASV12* overexpression	[90,133]

* stands for studies using Medaka.

## Data Availability

Not applicable.

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
