# Peer review of "Functional Validation of Osteoporosis Genetic Findings Using Small Fish Models"

_genes, 2022, doi:10.3390/genes13020279_

Round 1
Reviewer 1 Report
This article is interesting.
I request authors to add bioinformatics analysis of genes associated with Osteoporosis and show pathway analysis.
Also add a clear figure to show drug targets
Also add a section on natural compounds which have been investigated against Osteoporosis in fish models, etc
Author Response
This article is interesting.
RESPONSE:
We appreciate the Reviewer’s interest in our review and their constructive remarks.
I request authors to add bioinformatics analysis of genes associated with Osteoporosis and show pathway analysis.
RESPONSE:
To date, there are hundreds of “genes” (actually, loci/SNPs) associated with Osteoporosis (including BMD, fracture, heel ultrasound, bone structure), which make such a visualization challenging and problematic. The reader might be referred to specialized tools such as MSK-KP (Westendorf, J.J., Bonewald, L.F., Kiel, D.P. et al. The Musculoskeletal Knowledge Portal: improving access to multi-omics data. Nat Rev Rheumatol 18, 1–2 (2022). https://doi.org/10.1038/s41584-021-00711-1) which allow a meaningful pathway presentation. We have revisited the analysis performed by largest GWAS and we have now added a few lines on the pathway analysis from Morris et al. (2019) as it follows:
“Morris. et al performed pathway analysis for potential causal genes within 100kbp of the fine-mapped SNPs, identifying known pathways such as Wnt signaling, endochondral ossification, osteoclast and osteoblast signaling. Also, other pathways were identified such as DNA damage response, neural crest differentiation, mesoderm commitment, TGF-beta signaling, FTO obesity variant mechanism, transcription factor regulation of adipogenesis, estrogen signaling, BMP signaling and regulation[15]. Pathway analysis demonstrates the complex function of candidate genes and reinforces the need for functional studies to clarify causality.”
Also add a clear figure to show drug targets
RESPONSE:
Osteoporosis drug targets were recently reviewed1. Because our review has a focus on zebrafish for functional studies of the genetics of osteoporosis, we have given preference to this emerging model to be highlighted in pictures instead of known drug targets for osteoporosis.
Also add a section on natural compounds which have been investigated against Osteoporosis in fish models, etc
RESPONSE:
The use of natural compounds against Osteoporosis is a fascinating field that have been increasingly tested using the small-fish for pre-clinical modeling. However, this manuscript covers the genetics of Osteoporosis, thus we deem this section out-of-the-scope for this paper. To highlight this aspect of current research, we have added the following information to our text:
“It is worth mentioning that zebrafish have been increasingly used to test potential natural compounds for the treatment of osteoporosis. Natural compounds have been tested using glucocorticoid, glucose-induced osteoporosis (OP) and rankl-induced OP (medaka) to validate their anti-resorptive and anabolic activities [115].”
Reviewer 2 Report
The authors propose a review on how small fish such as zebra fish can be used to validate genetic hits of osteoporosis and low bone density. The manuscript is well detailed and walk the reader through the complexity of linking genetic mapping to molecular mechanisms. The authors review the strength and weakness of teleost fish to validate genetic impact of SNPs as well as propose original proposition of how fish could be more suitable for high-throughput screening of drugs and even discuss “aquarium-to-bedside” example of drug identification. The manuscript is well written and clear. The breadth of the manuscript could, however, be increased by addressing the following comments:
1- References should be added and described better for the interest sparked by the use of zebra fish on skull development and diploe formation as this model is unique and has several advantages over traditional model.
2- Table 1: Shen et al. ref 19 is the only reference mentioned without the date.
3- The zebra fish model is only introducing at the 6th page. It should come earlier in the text and in the introduction.
4- Line 105. The genetics of osteonecrosis of the jaw should also be discussed in resorptive therapies.
5- Figure 2C. The labelling is confusing. The mutant and control should be better identified.
6- A figure illustrating differences between zebrafish and mammal would be of great visual support. It should detail distinction in bone structure and site where endochondral ossification can be studied as fish do not posse long bone.
7- Line 191. Define ATAC-Seq.
8- Line 220. Need references.
9- Line 348 why is there a word in different colours there?
10- Line 486. Comment should be added on mononuclear osteoclast vs. polynucleated osteoclasts that are typically seen in mammals.
11- The authors should guide the reader in describing better trabecular vs. cortical bone development in fish and how parallel could be drawn in humans. Anti-osteoporotic drugs tend to act more at one of the 2 localizations depending on their mechanism of action.
12- Aging is described in the conclusion but could be earlier in the text as it is important to understand osteoporosis.
13- Use of bold and italic should be more consistent. Latin locution should be italicized. Line 311 for instance showed in silico bold and italicized.
Author Response
The authors propose a review on how small fish such as zebra fish can be used to validate genetic hits of osteoporosis and low bone density. The manuscript is well detailed and walk the reader through the complexity of linking genetic mapping to molecular mechanisms. The authors review the strength and weakness of teleost fish to validate genetic impact of SNPs as well as propose original proposition of how fish could be more suitable for high-throughput screening of drugs and even discuss “aquarium-to-bedside” example of drug identification. The manuscript is well written and clear.
RESPONSE:
We appreciate the Reviewer’s positive take on our review.
The breadth of the manuscript could, however, be increased by addressing the following comments:
1- References should be added and described better for the interest sparked by the use of zebra fish on skull development and diploe formation as this model is unique and has several advantages over traditional model.
RESPONSE:
To reinforce studies involving the zebrafish skull to model craniofacial diseases we have added a few lines to our text:
“In 2016, Kague et al took advantage of the transparency offered by zebrafish to show the complete development of the cranial sutures in vivo for the first time [87]. The same work demonstrated ectopic suture formation and the development of Wormian bones. (bones surrounded by sutures) in zebrafish lacking the transcription factor sp7. Remarkably, these findings in zebrafish precisely recapitulated those observed in a reported patient affected by osteogenesis imperfecta recessive that carried homozygous loss-of-function mutation in SP7 [87]. Therefore, zebrafish have efficiently modelled craniosynostosis, a premature fusion of the cranial sutures. These models are indicated in Table 2. Furthermore, zebrafish was recently used to validate new BMD candidate genes identified from the recent Skull-BMD-GWAS, providing functional evidence that Skull-BMD loci provide a repertoire of genes to be tested for involvement in craniofacial malformation and BMD regulation. Medina-Gomez et al. performed the largest skull-BMD-GWAS meta-analysis involving 43,800 individuals, identifying 59 loci, from which 4 loci were novel[5]. The group tested zic1, atp6v1c1a, atp6v1c1b, prkar1a and prkar1b in zebrafish, unprecedently showing that loss-of-function of each of these genes could lead to craniofacial malformation and abnormal BMD, thus, exemplifying the use of zebrafish to elucidate genes playing a role also in developmental craniofacial conditions.”
We have not added references regarding developmental studies of the early and larval zebrafish craniofacial skeleton because we consider this out of the scope of this specific review.
2- Table 1: Shen et al. ref 19 is the only reference mentioned without the date.
RESPONSE:
Thank you. We have amended our table.
3- The zebra fish model is only introducing at the 6th page. It should come earlier in the text and in the introduction.
RESPONSE:
Thank you. We have amended our introduction, adding ageing and briefly mentioning zebrafish.
4- Line 105. The genetics of osteonecrosis of the jaw should also be discussed in resorptive therapies.
RESPONSE:
We added the following text to our review:
“Bisphosphonate-induced osteonecrosis of the jaw (BRONJ) is also heritable2. Recently, whole-exome sequencing analyses resulted in a modest success of identifying variants associated with this adverse event3.”
5- Figure 2C. The labelling is confusing. The mutant and control should be better identified.
RESPONSE:
We have amended the figure to improve its interpretation (file enclosed).
6- A figure illustrating differences between zebrafish and mammal would be of great visual support. It should detail distinction in bone structure and site where endochondral ossification can be studied as fish do not posse long bone.
RESPONSE:
Thank you. We have added one more figure (current Figure 3, (file enclosed) following the suggestions of reviewer 2 about bone structure.
7- Line 191. Define ATAC-Seq.
RESPONSE:
Done.
8- Line 220. Need references.
RESPONSE:
We have added the following reference correspondent to this section.
“Slavica Dimitrieva, Philipp Bucher, UCNEbase—a database of ultraconserved non-coding elements and genomic regulatory blocks, Nucleic Acids Research, Volume 41, Issue D1, 2013, pp. D101–D109, https://doi.org/10.1093/nar/gks1092”
9- Line 348 why is there a word in different colours there?
RESPONSE:
Thank you, it was an oversight. It is now fixed.
10- Line 486. Comment should be added on mononuclear osteoclast vs. polynucleated osteoclasts that are typically seen in mammals.
RESPONSE:
We have added a few lines about mono- and multinucleated osteoclasts in mammals and their similarity with zebrafish as follows:
“In mammals osteoclast maturation involves multinucleation at a late phase of osteoclast differentiation. However, those osteoclasts that remain mononucleated keep their characteristic phenotype, with expression of osteoclast-related markers such as TRAP and Cathepsin K, and retain low levels of bone resorption ability 4. As in mammals, both mono- and multinucleated osteoclasts in zebrafish contribute to allometric bone growth and express osteoclast markers, such as TRAP 5. Zebrafish mononucleated osteoclasts are present from the second week post-fertilization, they are predominant in juvenile stages, and at thin skeletal elements (neural arches, nasal bones) of adults. Multinucleated osteoclasts are observed at around 40dpf and are the predominant osteoclast types of adults 5.”
11- The authors should guide the reader in describing better trabecular vs. cortical bone development in fish and how parallel could be drawn in humans. Anti-osteoporotic drugs tend to act more at one of the 2 localizations depending on their mechanism of action.
RESPONSE:
Thank you. We added the clause on importance of trabecular vs. cortical compartment in fish and how parallel could be drawn in humans:
“Their [trabeculations] specific development in fish have not yet been characterized. However, they develop after the complete formation of the vertebral column centrae and accompany the growth of the fish during life. As the fish grows in size, more trabeculations will be observed surrounding the vertebral centrae. A reduction in trabeculation in fish is attributed to the lower mechanical load of aquatic species in comparison with that of terrestrial. It is important to understand the differences between trabecular vs. cortical compartmentalization in fish and humans, since anti-osteoporotic drugs tend to act more at one compartment depending on their mechanism of action. “
12- Aging is described in the conclusion but could be earlier in the text as it is important to understand osteoporosis.
RESPONSE:
Thank you. We mentioned the age earlier in the text.
13- Use of bold and italic should be more consistent. Latin locution should be italicized. Line 311 for instance showed in silico bold and italicized.
RESPONSE:
Thank you. It is now fixed.
- Starling, S. New anti-osteoporosis drug target identified. Nature Reviews Endocrinology 17, 4-5 (2021).
- Lee, K.H. et al. Identifying genetic variants underlying medication-induced osteonecrosis of the jaw in cancer and osteoporosis: a case control study. Journal of Translational Medicine 17, 381 (2019).
- Yang, G. et al. Pharmacogenomics of osteonecrosis of the jaw. Bone 124, 75-82 (2019).
- Kodama, J. & Kaito, T. Osteoclast Multinucleation: Review of Current Literature. International Journal of Molecular Sciences 21, 5685 (2020).
- Witten, P.E., Hansen, A. & Hall, B.K. Features of mono- and multinucleated bone resorbing cells of the zebrafish Danio rerio and their contribution to skeletal development, remodeling, and growth. J Morphol 250, 197-207 (2001).
Round 2
Reviewer 1 Report
I reject this paper as authors have not addressed my comments in a serious manner.
Author Response
Reviewer 1 comments
“I request authors to add bioinformatics analysis of genes associated with Osteoporosis and show pathway analysis.”
RESPONSE:
<< We have not performed new bioinformatic analyses but we have added bioinformatic analysis from the largest GWAS (Morris et al. 2019). In the largest GWAS pathway analysis was performed, identifying core pathways, such as those associated with skeletogenesis and WNT signalling, but also new pathways as neural crest and obesity. We have added the following information to our text:
“Morris et al. performed pathway analysis for potential causal genes within 100kbp of the top SNPs, identifying known pathways such as Wnt signalling, endochondral ossification, osteoclast and osteoblast signalling. Also, other pathways were identified such as DNA damage response, neural crest differentiation, mesoderm commitment, TGF-beta signalling, FTO obesity variant mechanism, transcription factor regulation of adipogenesis, estrogen signalling, and BMP signalling and regulation [17]. Pathway analysis demonstrates the complex function of candidate genes and reinforces the need for functional studies to clarify causality. >>
Also add a clear figure to show drug targets
RESPONSE:
<<We have added a Figure showing drug targets and therapeutics for OP. >>
Also add a section on natural compounds which have been investigated against Osteoporosis in fish models, etc
RESPONSE:
<<We have expanded our text and incorporated a few examples of works analysing natural compounds in zebrafish. Our text now reads:
“Fish models of induced OP have been recently reviewed [158]. It is worth mentioning that zebrafish have been increasingly used to test potential natural compounds for the treatment of osteoporosis. Because small fish offer a rapid and non-invasive in vivo platform for bone formation, the model has been used to screen a variety of natural compounds derived from traditional herbal medicine, especially Chinese medicine, and to validate in vitro findings. Natural compounds have been tested using glucocorticoid, glucose-induced OP and rankl-induced OP (medaka) to validate their anti-resorptive and anabolic activities [158]. For example, liquiritigenin (a flavonoid extracted from Glycyrrhiza glabra roots) has a positive effect in osteoblast differentiation, as well as to halt osteoclast activity in vitro. Carnovali et al. tested liquiritigenin in zebrafish, revealing increased bone formation rates in a dose dependent manner and reduced TRAP staining in zebrafish scales of glucocorticoid induced OP [159]. A similar approach was used to analyse natural foods with antioxidant properties (curcuma, papain, bromelain and black pepper), which showed anabolic effects during development and reduced osteoclastic activity in zebrafish scales [160]. Zebrafish have been of interest to also test marine alkaloids [161], which altogether shall boost the discovery of potential alternative therapeutics.” >>